# Poxvirus Infections in Dairy Farms and Transhumance Cattle Herds in Nigeria

**DOI:** 10.3390/v15051051

**Published:** 2023-04-25

**Authors:** David Oludare Omoniwa, Irene Kasindi Meki, Caleb Ayuba Kudi, Anthony Kojo Sackey, Maryam Aminu, Adeyinka Jeremy Adedeji, Clement Adebajo Meseko, Pam Dachung Luka, Olayinka Oluwafemi Asala, Jolly Amoche Adole, Rebecca Bitiyong Atai, Yakubu Joel Atuman, Tirumala Bharani Kumar Settypalli, Giovanni Cattoli, Charles Euloge Lamien

**Affiliations:** 1Department of Veterinary Medicine, Surgery and Radiology, University of Jos, Jos 930001, Plateau State, Nigeria; omoniwad@unijos.edu.ng; 2Department of Veterinary Medicine, Ahmadu Bello University, Zaria 810211, Kaduna State, Nigeria; cakudi@abu.edu.ng (C.A.K.);; 3Animal Production and Health Laboratory, Animal Production and Health Section, Joint FAO/IAEA Division, Department of Nuclear Sciences and Applications, International Atomic Energy Agency, P.O. Box 100, 1400 Vienna, Austria; i.meki@iaea.org (I.K.M.); t.b.k.settypalli@iaea.org (T.B.K.S.); g.cattoli@iaea.org (G.C.); c.lamien@iaea.org (C.E.L.); 4Department of Microbiology, Ahmadu Bello University, Zaria 810211, Kaduna State, Nigeria; 5National Veterinary Research Institute, Vom 930103, Plateau State, Nigeria; adeyinka.adedeji@nvri.gov.ng (A.J.A.); olayinka.asala@nvri.gov.ng (O.O.A.); rebecca.bityong@nvri.gov.ng (R.B.A.); yakubu.atuman@nvri.gov.ng (Y.J.A.)

**Keywords:** poxvirus, dairy farms, transhumances, cattle, co-infection, pseudocowpox, HRM assay, Nigeria

## Abstract

Lumpy Skin disease (LSD) is an economically important disease in cattle caused by the LSD virus (LSDV) of the genus *Capripoxvirus*, while pseudocowpox (PCP) is a widely distributed zoonotic cattle disease caused by the PCP virus (PCPV) of the genus *Parapoxvirus*. Though both viral pox infections are reportedly present in Nigeria, similarities in their clinical presentation and limited access to laboratories often lead to misdiagnosis in the field. This study investigated suspected LSD outbreaks in organized and transhumance cattle herds in Nigeria in 2020. A total of 42 scab/skin biopsy samples were collected from 16 outbreaks of suspected LSD in five northern States of Nigeria. The samples were analyzed using a high-resolution multiplex melting (HRM) assay to differentiate poxviruses belonging to *Orthopoxvirus*, *Capripoxvirus*, and *Parapoxvirus* genera. LSDV was characterized using four gene segments, namely the RNA polymerase 30 kDa subunit (RPO30), G-protein-coupled receptor (GPCR), the extracellular enveloped virus (EEV) glycoprotein and CaPV homolog of the variola virus B22R. Likewise, the partial B2L gene of PCPV was also analyzed. Nineteen samples (45.2%) were positive according to the HRM assay for LSDV, and five (11.9%) were co-infected with LSDV and PCPV. The multiple sequence alignments of the GPCR, EEV, and B22R showed 100% similarity among the Nigerian LSDV samples, unlike the RPO30 phylogeny, which showed two clusters. Some of the Nigerian LSDVs clustered within LSDV SG II were with commonly circulating LSDV field isolates in Africa, the Middle East, and Europe, while the remaining Nigerian LSDVs produced a unique sub-group. The B2L sequences of Nigerian PCPVs were 100% identical and clustered within the PCPV group containing cattle/Reindeer isolates, close to PCPVs from Zambia and Botswana. The results show the diversity of Nigerian LSDV strains. This paper also reports the first documented co-infection of LSDV and PCPV in Nigeria.

## 1. Introduction

Lumpy skin disease (LSD) is a viral disease affecting cattle and water buffaloes. LSD is a transboundary disease with significant economic importance via the losses encountered by livestock owners [1,2,3]. LSD is caused by the lumpy skin disease virus (LSDV): a double-stranded DNA virus containing around 150 kilobase pairs (kbp) and a genome enclosed in a lipid envelope. LSDV belongs to the genus *Capripoxvirus* (CaPV) within the family *Poxviridae* [4,5]. In addition to LSDV, the sheeppox virus (SPPV) and goatpox virus (GTPV), two antigenically related viruses, also belong to the same genus [6,7]. Cattle of all breeds and ages can be affected, with young and lactating animals being particularly susceptible [8]. LSD morbidity is around 10–14%; however, a higher morbidity of 40% and sometimes even 100% has been reported in non-vaccinated dairy and feedlot animals [4,9,10]. The mortality rate ranges from 1 to 10% in mild infections, but case fatality rates can occasionally reach 50–75% [11,12,13,14,15]. Historically, the earliest report of LSD was in 1929 in Zambia, where it spread across the African continent [16,17]. Nigeria’s first report of LSD was in the 1970s [18,19]. Since then, reports of outbreaks have been consistent in transhumance herds and established cattle farms [1,11,20].

The transmission of LSDV is primarily caused by blood-sucking insects and ticks [21,22,23,24]. It has also been demonstrated experimentally to occur by close indirect contact [25], through fomites such as contaminated feed and equipment, or directly through semen or vertical transmission from a dam to a calf [26,27,28,29].

LSD symptoms in cattle vary from mild to severe. The most typical clinical signs are fever (40–41 °C); the appearance of multiple skin nodules that progress to sitfast lesions covering the neck, back, perineum, tail, limbs, and genital organs, and mucous membranes. Lesions may also involve subcutaneous tissues, musculature, and internal organs [28,30]. Affected animals can also exhibit lameness, emaciation, and the cessation of milk production. Edema of the limbs, brisket, and lymphadenitis are highly prominent; sometimes, affected animals may die. In addition, pneumonia is a typical sequel in animals with lesions in the mouth and respiratory tract [31,32].

Severe cases of LSD are highly characteristic and easy to recognize. However, early stages of infection and mild cases may be challenging to diagnose, thus requiring laboratory confirmation. The following diseases are considered differentials of LSD: pseudo lumpy skin disease/bovine herpes mammillitis, bovine papular stomatitis, insect bites, urticaria, photosensitization, and pseudocowpox. Pseudocowpox is a worldwide zoonotic cattle disease caused by the pseudocowpox virus (PCPV) of the genus *Parapoxvirus*, sub-family Chordopoxvirinae, within the family *Poxviridae* [33]. Freshly calved and recently introduced adult cattle are the most susceptible groups to the disease.

The transmission of parapoxvirus infection occurs through damaged or broken skin and sometimes through the oral mucosa [34]. Virus replication near the port of entry is accompanied by a well-characterized clinical course that progresses through the stages of macules, papules, vesicles, pustules, and scabs [35]. The lesions of pseudocowpox are usually confined to cattle teats. Nevertheless, they may also develop on muzzles and in the mouths of nursing calves. Symptoms include ring-shaped or horseshoe-shaped scabs on the teats, which usually heal within six weeks [36]. The pseudocowpox virus is zoonotic and causes a self-limiting disease known as Milker’s nodules. This disease is considered an occupational hazard to individuals who work in the dairy industry and come in contact with infected animals that shed the virus in saliva, nasal secretions, and lesions over the udder, trunk, and limbs [37,38]. It may be transmitted by the direct or indirect contact of lesions. In humans, it is characterized by nodular lesions on the hands, forearms, and sometimes on the face. Pseudocowpox was reported in Nigeria in the early 1980s [39]. Co-infection by viruses of the family *Poxviridae*, for instance, Parapoxviruses (PCPV) with Orthopoxviruses (vaccinia virus VACV) or Capripoxviruses (GTPV), has been previously reported [35,40,41]. Although PCPV and other parapoxviruses of cattle are known to occur worldwide, there is little molecular evidence and sequence information regarding PCPV in Africa. Recently, Zambia and Botswana have described molecular PCPV for the first time [42,43]. However, in Nigeria, no molecular information is available. Several molecular-based approaches, such as real-time PCR or probe-based PCR and high-resolution melting (HRM) analysis, have been developed to differentiate and genotype poxviruses [44]. Therefore, our study aimed to characterize co-infecting poxviruses in samples collected from suspected LSD-infected animals in dairy and transhumance cattle herds in Nigeria.

## 2. Materials and Methods

### 2.1. Study Area and Sample Collection

Cattle with pox-like lesions were sampled in 2020. The samples were collected from five northern states of Nigeria: Bauchi, Kaduna, Nasarawa, Niger, and Plateau states (Figure 1). Some states had a few integrated/dairy farms where animals were kept within a confined and controlled environment (farms). In other states, herds were kept in a semi-intensive system by individual community members, and they moved in and out of confinement to graze. Samples were collected from animals in both integrated farms and semi-intensive system farms.

Scab lesions were collected from 42 cattle presenting suspected clinical signs of the poxvirus infection. The skin samples, no more than 2 g, were collected from each animal and placed in a labeled sterile universal tube. The collected samples were shipped at +4 °C to the Virology Division of the National Veterinary Research Institute, Vom, and stored at −20 °C until laboratory diagnosis.

### 2.2. Detection of Lumpy Skin Disease Virus by the Polymerase Chain Reaction

The samples were homogenized to 20% in a phosphate-buffered saline (PBS) solution. The total DNA was extracted using the QIAamp DNA Mini kit (QIAGEN, Hilden, Germany) following the manufacturer’s instructions. A previously described conventional PCR was employed to detect the presence of CaPVs in the collected samples [45].

DNA extracts from nineteen PCR-positive samples were prepared and sent to the Animal Production and Health Laboratory, Joint FAO/IAEA Laboratories, Seibersdorf, Austria, for confirmation and sequencing. The samples were further confirmed as CaPV positive using a real-time PCR, as described by Bowden et al. [46].

To determine the cross-infection of ortho- and parapoxviruses with capripoxvirus-positive samples, a high-resolution multiplex melting (HRM) assay developed by Gelaye and co-workers [44] was employed. This assay can detect and differentiate the co-infection of poxviruses belonging to *Orthopoxvirus*, *Capripoxvirus*, and *Parapoxvirus* genera based on the amplicon melting temperature and profile. Briefly, the PCR mixture was set up in a 20 μL reaction volume containing 1× of SsoFast EvaGreen Supermix (Bio-Rad, USA), 200 nM each of the forward and reverse primers for each genus, and 2 μL of template DNA. The PCR was performed in a CFX96 PCR machine, and the program consisted of an initial denaturation step at 95 °C for 4 min, followed by 40 cycles at 95 °C for 1 s, 59 °C for 2 s, and 70 °C for 2 s with fluorescence reading at the end, followed by heating at 95 °C for 30 s, and cooling at 65 °C for 1 min, and melting from 65 °C to 85 °C at 10 s/0.2 °C with a fluorescence reading at each °C, and then 37 °C for 1 min. The amplification plots were analyzed using the Bio-Rad CFX Maestro^TM^ Software version 1.1, and the HRM profiles were analyzed using the Precision Melt Analysis^TM^ Software version 1.2 (Bio-Rad, Hercules, CA, USA).

### 2.3. Amplification and Sequencing of Selected Poxvirus Genes

For the further genetic characterization of the poxviruses positive samples, four CaPV genes: the RNA polymerase 30 kDa subunit (RPO30), G-protein-coupled receptor (GPCR), extracellular enveloped virus (EEV) glycoprotein, and CaPV homolog of the variola virus B22R, and a partial B2L gene for PPVs was amplified following the conditions previously described [47,48,49]. The PCR products were separated by electrophoresis on a 2% agarose gel and visualized using a Gel Documentation System (Bio-Rad). The PCR products were sent for sequencing using both forward and reverse primers at LGC Genomics (Berlin, Germany).

### 2.4. Sequence and Phylogenetic Analysis

Vector NTI software (Invitrogen) version 11.5 was used to check the raw data quality and assemble the sequences. The sequences for each targeted gene were aligned with reference CaPV or PPV sequences retrieved from GenBank, using the Muscle algorithm and the codon option in MEGA 7 v7.0.26 [50]. The multiple sequence alignments of partial EEV and B22R were performed using BioEdit (v7.2.6). Bayesian phylogenetic inference of the complete RPO30, GPCR, and partial B2L was performed using BEAST v1.8.4. First, the aligned sequence files were converted to a Nexus format on Seaview software. The Markov-chain Monte Carlo (MCMC) log files generated by BEAST were further inspected using the TRACER v1.7.1 program to determine the optimum burn-in value number based on the effective sample sizes (ESS) values. Finally, maximum Clade Credibility (MCC) trees were generated using a TreeAnnotator [51]. The trees were visualized and annotated using the interactive tree of life (iTOL) [52].

## 3. Results

### 3.1. Clinical Signs and Symptoms of the LSD Outbreak

The sampled animals showed the following clinical signs: pyrexia, skin nodules/scabs, lymphadenopathy lameness, and hyper salivation (Figure 2). The overall morbidity rate ranged from 8.24% to 36.17%, while the mortality rate was 0 to 7.32%. The highest morbidity and mortality rates were recorded in Friesian (36.17% and 4.26%) and Simmental breeds (22.76% and 7.32%) (Table 1).

### 3.2. Molecular Detection and Diagnosis of Poxviruses

A total of 19 (45.2%) samples were positive for CaPV by conventional PCR and were confirmed by a real-time PCR [45,46]. Further analyses using the HRM assay showed that the samples were LSDV positive, and five were co-infected with the pseudocowpox virus (PCPV) (Table 1).

### 3.3. LSDV Gene Amplification and Sequencing

Eleven of the 19 samples were amplified and sequenced successfully for the four targeted CaPV genes (RPO30, GPCR, EEV, and B22R). Likewise, the PPV partial B2L gene of two co-infected samples was sequenced. After the quality check and editing, the final sequences were deposited in GenBank under the accession numbers OQ094248 to OQ094258 (RPO30 gene), OQ094259 to OQ094269 (GPCR gene), OQ094270 to OQ094280 (partial EEV glycoprotein), OQ094281 to OQ094291 (partial B22R), and OQ094292 to OQ094293 (partial B2L gene).

### 3.4. Sequence Analysis

The multiple sequence alignments of the GPCR, EEV, and B22R genes showed a 100% similarity among the Nigerian LSDV samples of this study. These sequences were also identical to those of an LSDV isolate collected in Nigeria in 2018 (OK318001) [53]. By contrast, sequence alignment using the RPO30 gene showed two clusters of the samples. Nevertheless, further gene alignment analysis showed a 27-nucleotide insertion (175–201) in the EEV glycoprotein of all Nigerian LSDVs, similar to the LSDV field isolates from Turkey (MN995838), Serbia (KY702007) and Russia (MH893760). This insertion distinguishes them from LSDV Neethling isolates, Neethling-derived vaccines, LSDV isolates from China, and recombinant LSDVs from Russia (Figure 3a). Furthermore, multiple sequence alignments of the targeted B22R fragment reiterated not only the sequence similarity of the Nigerian LSDVs to the field isolates mentioned above but also showed the missing single nucleotide insertion at positions 102 and 745 in the LSDV Neethling and the LSDV KSGPO-240 vaccine strains, respectively (Figure 3b).

Phylogenetic analysis of the RPO30 gene showed five Nigerian LSDVs clustered within LSDV SG II together with the commonly circulating LSDV field isolates in Africa, LSDV Sudan (GU119938), the Middle East; Egypt (GU119947), Europe, LSDV Serbia (KY702007), as well as the newly emerged LSDV variants from Russia (MH893760). The remaining six LSDVs from Nigeria did not cluster with the existing LSDV sub-groups but formed a sub-cluster called the SG-Nig group (Figure 4).

Sequence analysis of the GPCR gene clustered all Nigerian LSDVs in the same sub-group (SGI) together with the LSDV field isolates, notwithstanding the presence or absence of a 12-nucleotide deletion of the GPCR gene (Figure 5). Moreover, this deletion distinguished Nigeria LSDV isolates from the LSDVs of Bangladesh, India, China, and the Kenyan isolates, NI-2490 (AF325528) and LSDV Kenya (MN072619), which were also grouped in SG I.

The analyzed B2L gene sequences of PCPV co-infected Nigerian samples were 100% identical. They differed from those in GenBank, with a 99.77% and 99.32% nucleotide similarity to PCPVs from Zambia (MT448679) and Botswana (MW748473), respectively. The phylogenetic analysis based on the partial B2L gene clustered the Nigerian PCPV samples within the PCPV group of cattle/reindeer isolates and with those from Zambia and Botswana (Figure 6).

## 4. Discussion

The earliest Nigerian reports of LSD and pseudocowpox were by Woods [19] and Asagba in the 1970s [18]. However, since then, the lack of information on these diseases, especially in the West African sub-region, has persisted. In 2020, sporadic LSD outbreaks were reported in the Kano, Plateau, and Nasarawa states of Nigeria [1,18,20,30], with the affected cattle showing a typical clinical presentation of LSD [17]. Samples were collected from outbreaks in transhumance, integrated, and dairy farms and were confirmed for the presence of LSDV. A total of 19 LSDV-positive samples (45.2%) were analyzed by the HRM assay, which differentiated eight poxviruses and five samples from four states, namely: Bauchi, Nasarawa, Kaduna, and Niger states, which were positive for co-infections of LSD and pseudocowpox. Despite this, all analyzed samples were collected from animals presenting clinical signs, and 54.8% were negative by the conventional PCR that targeted the CaPV GPCR gene, implying that these animals were probably infected with other poxviruses [45].

The discovery of co-infections in this study supports earlier reports suggesting that PCPV coinfects with other parapoxviruses [40,54]. The morbidity and mortality rates per herd observed in Nigeria LSD outbreaks were similar to those recorded in other LSD outbreaks, such as in Ethiopia (21.2% and 4.5%), Zimbabwe (30.95% and 8.77%), and Sudan (4.1% and 2.4%) [55,56,57]. Moreover, the high morbidity rate observed in Friesian cattle in Nigeria (36.17%) has also been reported by imported Friesian (37.9%) in Sudan and Oman [33,58]. Another observation from this study was that the LSD outbreak in PCPV-co-infected herds had more severe clinical signs and symptoms and relatively higher morbidity rates than in LSDV-infected herds. LSD is endemic in Nigeria [1,18,20,30], and pseudocowpox occurs worldwide [34]. However, only one report of the presence of pseudocowpox in Nigeria could be found. This implies either under-reporting, misdiagnosis in place of other infections, or unawareness of the disease among farmers, animal healthcare workers, and veterinarians. There is also a lack of capability to differentiate using molecular techniques and cases that present pox-like lesions, especially in the LSD-endemic regions of Africa, with most cases being summarily treated in the field.

The advent of the HRM assay provides a reliable and accurate test to differentiate poxviruses and improve diagnostic accuracy. As the initial purpose of this work was to analyze LSDV in transhumance herds, the presence of co-infection with PCPV in these herds could have been overseen if the HRM screening tool was not available for this study [44]. This emphasizes the need to use differential diagnosis that can detect co-infection, even when the direct detection method identifies a primary pathogen [59,60,61]. The application of such tools has previously enabled the identification of PCPV in LSD suspicions in Zambia and Botswana, revealing, for the first time, the presence of these pathogens using molecular tools and genomic approaches [42,43]. In Brazil, the co-infection of poxviruses such as VACV and Orf virus (ORFV), VACV and PCPV, or even the triple infection of the bovine papular stomatitis virus (BPSV), PCPV, and VACV has been reported in Bovine based on serological and molecular methods [35,40,62]. A co-infection of the camelpox virus (CMLV) and PCPV has also been shown to occur in Camels in Ethiopia [44].

Sequence analysis of the Nigeria samples showed the slight diversity of LSDV isolates in this country depending on the gene used for analysis, indicating the great importance of the surveillance and monitoring of LSDV by country. For instance, the RPO30 phylogenetic tree showed that LSDV isolates from Nigeria belonged to two different sub-groups: sub-group SG II and a new subgroup formed only by the Nigerian isolates (SG Nig). However, the GPCR phylogenetic analysis clustered all the Nigeria LSDV isolates in the same sub-group (SG I), which was reiterated by the multiple sequence alignments of the partial EEV and B22R genes [45,63,64,65]. Although the B2L sequences of the 2 Nigerian PCPV isolates were different from those of other PCPVs available in public databases, they phylogenetically clustered close to those from Zambia and Botswana within the cattle/reindeer isolates group [42,43].

The major finding of our study included the fact that the Nigerian LSDV isolates were similar to other known African isolates from Sudan, Ethiopia, South Africa, and Egypt, and European isolates from Bulgaria, Greece, Serbia, Turkey, and isolates from Russia, Kazakhstan, Iraq, and Israel, but different from recombinant LSDV field isolates from China, Thailand, Taiwan, and Russia [66]. On the other end, the Nigerian PCPV isolates were clustered with the Zambian and Botswana isolates.

The diversity of these two viruses could be primarily due to the natural evolution of viruses in an endemic area [67]. Other factors that could have aided the spread of the disease in Nigeria include the introduction of cattle into Nigeria from other countries, the movement of transhumance farmers across national boundaries and along cattle trade routes in Africa with little or no supervision, or coming into contact with other herds who are also on diverse journeys across Africa [68]. Political instability, war, and climate change, including drought and flooding, cause cattle dispersal all over Africa in search of greener pastures which help in spreading and introducing diseases across the continent. All these complicate disease surveillance and control in Nigeria and the African continent.

## 5. Conclusions

In conclusion, this study has highlighted the first documented co-infection of LSD and pseudocowpox in Nigeria, aided by the use of HRM assay, and has also shown the diversity of both LSD and pseudocowpox strains within the country. This study reiterates the need for the continuous surveillance and monitoring of LSDV and PCPV within Nigeria and the West African subregion in order to generate accurate epidemiological data as well as to help formulate crucial control strategies for disease outbreaks at both national and regional levels. It also demonstrates the power of differential diagnosis to identify causative agents during disease outbreaks accurately.

## Figures and Tables

**Figure 1 viruses-15-01051-f001:**
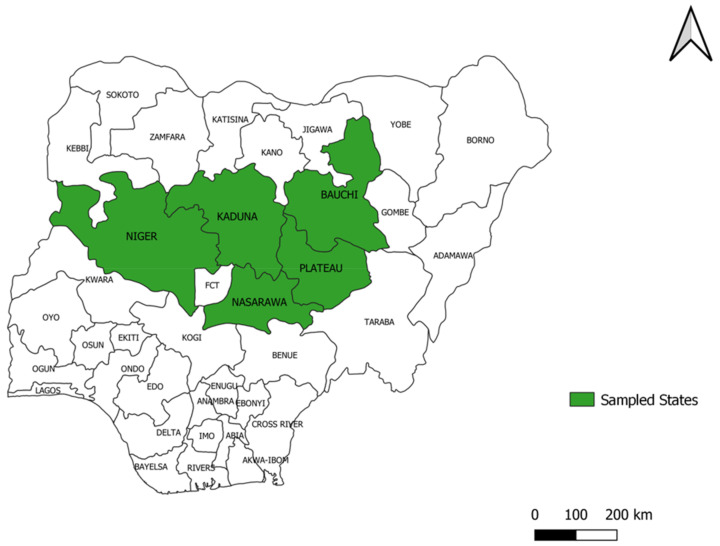
Map of Nigeria showing states where samples were collected.

**Figure 2 viruses-15-01051-f002:**
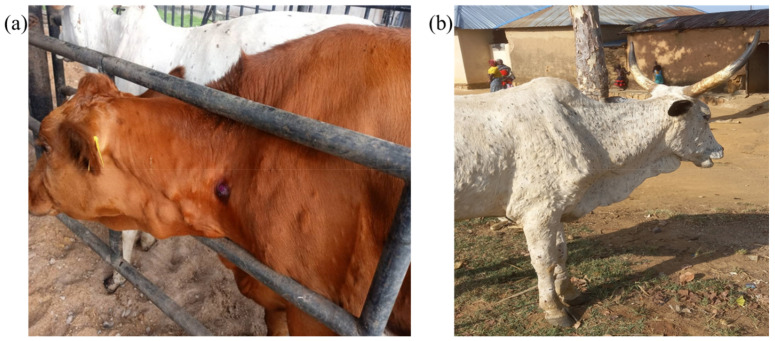
(**a**) LSD nodules on the neck of a Simmental cow in a dairy farm in Minna Niger State Nigeria and (**b**) Generalized LSD nodular skin lesion on a white Fulani cow in a transhumance herd in Vom, Plateau State, Nigeria.

**Figure 3 viruses-15-01051-f003:**
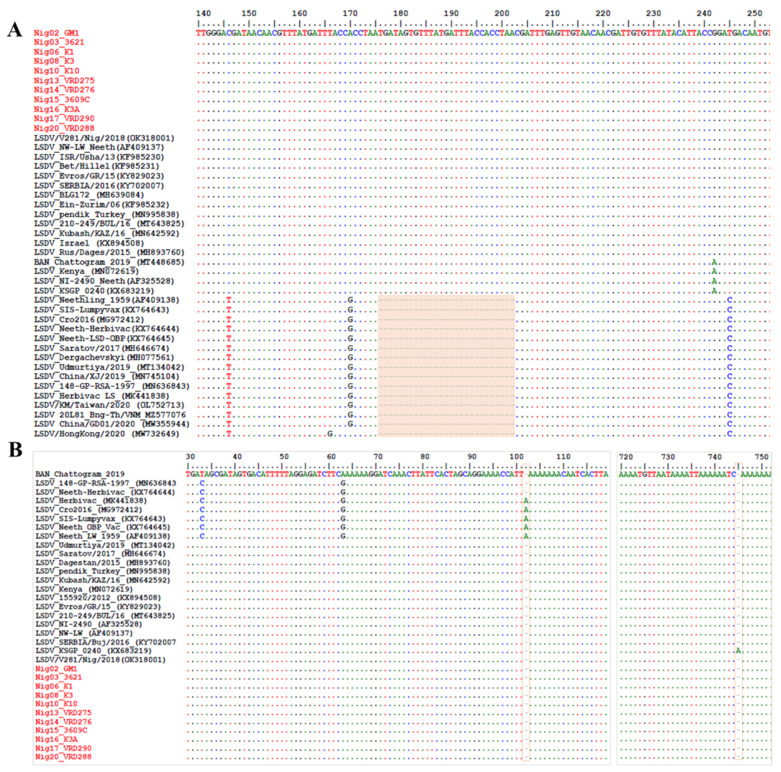
Multiple sequence alignment of the nucleic acid sequence of the (**A**) EEV glycoprotein gene and (**B**) B22R gene sequences of the Nigerian LSDV isolates aligned with representative LSDV sequences retrieved from GenBank. Nigeria LSDV isolates are in red, and the dots indicate the identical nucleotides in the alignment.

**Figure 4 viruses-15-01051-f004:**
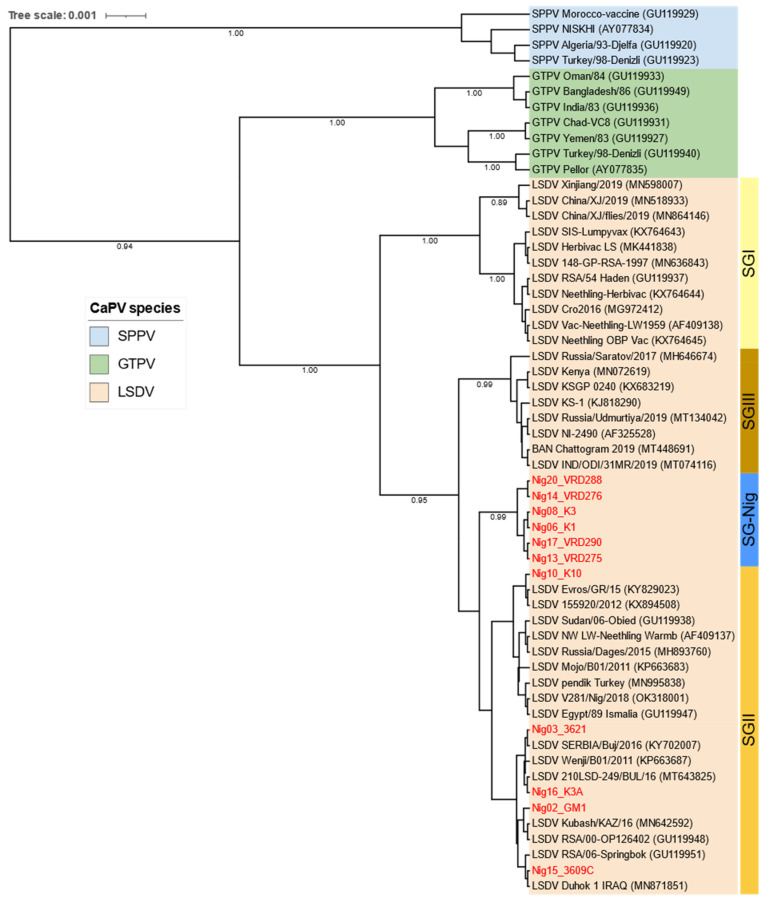
Maximum clade credibility (MCC) tree based on the complete RPO30 gene sequences of CaPVs, applying the HKY model and gamma-rate distributions, with LSDVs from Nigeria in red, visualized on iTOL.

**Figure 5 viruses-15-01051-f005:**
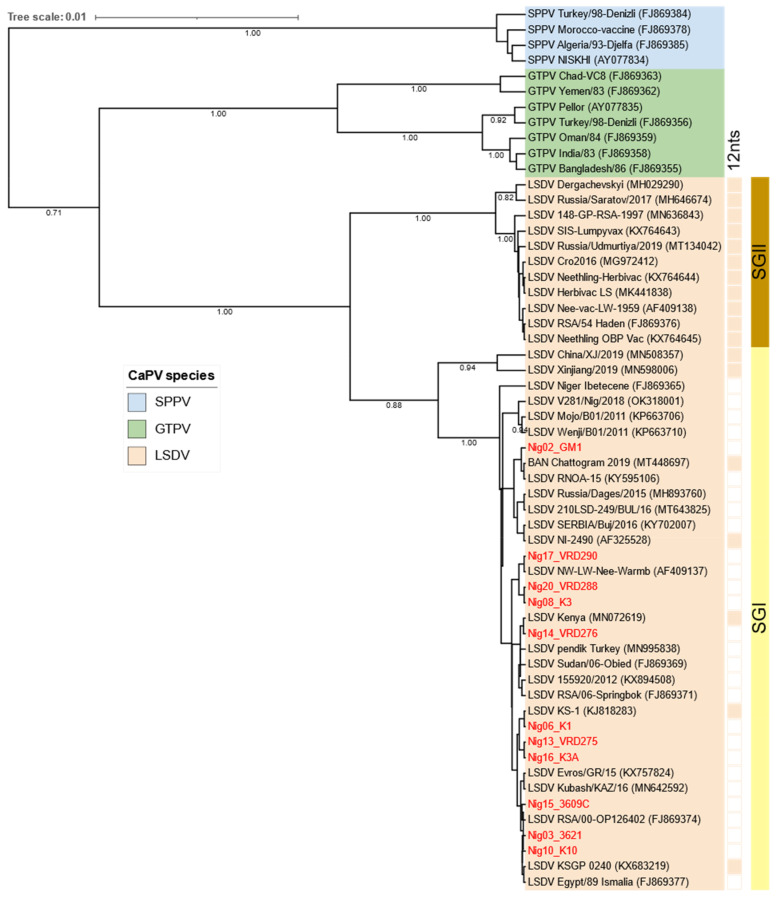
Maximum clade credibility (MCC) tree based on the complete GPCR gene sequences of CaPVs applying the HKY model and gamma-rate distributions visualized on iTOL with isolates clustering based on the presence (filled box) or absence (empty box) of a 12-nucleotide insertion in the GPCR gene, with LSDVs from Nigeria in red.

**Figure 6 viruses-15-01051-f006:**
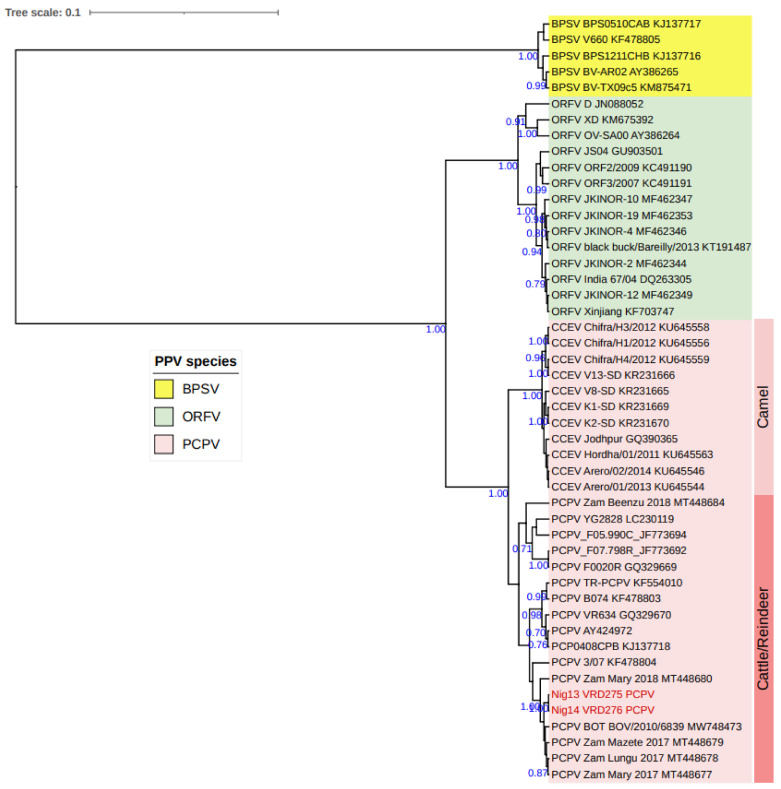
Maximum clade credibility (MCC) tree based on partial B2L gene sequences of PPVs applying the HKY model and gamma-rate distributions, with PCPV samples from Nigeria in red, visualized on iTOL.

**Table 1 viruses-15-01051-t001:** LSD suspected samples used in this study (*n* = 19): Sample names, collection area, host information and molecular detection data are presented.

Sample ID	Species/Breed	Location	Clinical Signs	Herd Size/ Morbidity/ Mortality	Age	Date of Collection	LSD Vaccination History	Cq (Bowden et al., 2008 [46])	Cq (Gelaye et al., 2017 [44])	Tm HRM	Remarks
VSD278	Bovine/Simmental	Minna Niger State	Pyrexia, skin nodules, enlarged lymph nodes	123/28/9	2 yr	18-08-2020	None	30.92	N/A	N/A	N/A
GM1	Bovine/White Fulani	Kanam Plateau State	Pyrexia, ocular discharge, nasal discharge, generalized skin nodules	57/8/0	9 months	22-09-2020	None	19.76	19.21	77.60	LSDV
3621	Bovine/Friesian	Vom Plateau State	Oral ulcers, pyrexia, lameness, generalized skin nodules, corneal opacity	47/17/2	1 yr	07-09-2020	None	26.31	24.83	77.60	LSDV
3600A	Bovine/Friesian	Vom Plateau State	Pyrexia, hypersalivation, enlarged lymph nodes, ocular discharge, generalized skin nodules	26/5/0	15 months	12-09-2020	None	32.15	30.45	77.60	LSDV
3600B	Bovine/Friesian	Vom Plateau State	Enlarged lymph nodes, ocular discharge, generalized skin nodules	26/5/0	15 months	12-09-2020	None	32.26	31.97	76.60	LSDV
K1	Bovine/White Fulani	Keana, Nasarawa state	Localized skin nodules, enlarged lymph nodes,	72/11/0	3 yr	22-08-2020	None	27.99	26.5	77.60	LSDV
K2	Bovine/White Fulani	Keana, Nasarawa state	Pyrexia, skin nodules, enlarged lymph nodes	72/11/0	9 yr	22-08-2020	None	26.57	25.43	77.60	LSDV
K3	Bovine/White Fulani	Keana, Nasarawa state	Generalized skin nodules, pyrexia, lameness	55/6/0	3 yr	22-08-2020	None	17.06	16.36	77.4 & 81.2	LSDV & PCPV
K9	Bovine/White Fulani	Keana, Nasarawa state	Localized skin nodules, enlarged lymph nodes, lameness	109/20/2	5 yr	22-08-2020	None	28.99	N/A	N/A	N/A
K10	Bovine/White Fulani	Keana, Nasarawa state	Ocular discharge, skin nodules, enlarged lymph nodes	109/20/2	1 yr	22-08-2020	None	19.91	19.5	77.20	LSDV
K12	Bovine/White Fulani	Keana, Nasarawa state	Pyrexia, Skin nodules, enlarged lymph nodes	109/20/2	3 yr	22-08-2020	None	24.90	16.54	N/A	N/A
VRD275	Bovine/White Fulani	Minna Niger State	Generalized skin nodules, pyrexia, lameness, enlarged lymph nodes, ocular discharge	216/23/5	2 yr	27-07-2020	None	21.97	20.41	77.4 & 81.2	LSDV & PCPV
VRD276	Bovine/Simmental	Minna Niger State	Generalized skin nodules, pyrexia, lameness, enlarged lymph nodes, nasal discharge, oral ulcers	123/28/9	4 yr	18-08-2020	None	19.59	17.74	77.4 & 81.2	LSDV & PCPV
3609C	Bovine/Friesian	Vom Plateau State	Generalized skin nodules, pyrexia, lameness, enlarged lymph nodes, nasal discharge, oral ulcers	47/17/2	10 months	13-09-2020	None	26.46	25.54	77.40	LSDV
K3A	Bovine/Sokoto Gudali	Keana, Nasarawa state	Pyrexia, Skin nodules, enlarged lymph nodes	55/6/0	4 yr	22-08-2020	None	21.09	20.31	77.20	LSDV
VRD290	Bovine/White Fulani	Tudun Wada Kaduna State	Generalized skin nodules, pyrexia, lameness, enlarged lymph nodes, ocular discharge	73/13/3	2 yr	16-08-2020	None	14.87	15.26	77.60	LSDV
Bau6	Bovine/White Fulani	Bauchi, Bauchi State	Pyrexia, hypersalivation, enlarged lymph nodes, ocular discharge, generalized skin nodules	85/7/2	7 yr	11-10-2020	None	31.40	30.34	76.6 & 81.4	LSDV & PCPV
VRD272	Bovine/White Fulani	Kara Plateau State	Generalized skin nodules, ocular discharge, pyrexia lameness, salivation.	78/7/0	3 yr	21-08-2020	None	31.41	18.2	81.40	LSDV
VRD288	Bovine/White Fulani	Tudun Wada Kaduna State	Pyrexia, skin nodules, enlarged lymph nodes	136/17/3	4 yr	05-07-2020	None	25.26	24.35	77.6 & 81.4	LSDV & PCPV

## Data Availability

The DNA sequences generated and used in the analysis for this study are available in the Gen-Bank under the accession numbers OQ094248 to OQ094293.

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
