# Peer review of "Poxvirus Infections in Dairy Farms and Transhumance Cattle Herds in Nigeria"

_viruses, 2023, doi:10.3390/v15051051_

Round 1

Reviewer 1 Report

Pleaser find comments in the uploaded document.

Author Response

The manuscript has been revised based on the suggestions/comments from the pdf file

Reviewer 2 Report

In the manuscript entitled “Poxvirus infections in dairy farms and transhumance cattle herds in Nigeria”, the authors described the detection and characterization of LSDV using molecular techniques, in Nigeria as well as the co-identification of Pseudo-cowpox in the same samples. The manuscript follows a logical outline and is recommended for publications. There are some concerns and questions that the authors should consider and address, prior to publication. Suggestions and comments are listed below.

Q1. The authors describe the genetic characterization and phylogenetic relatedness of field samples obtained from Nigeria, but omits to include a previous full genome sequence (OK318001) of isolate LSDV/V281/Nigeria/2018 to the analysis. Similarly, could the sequences from LSDV outbreaks in Uganda 2017 – 2018 be included in the analysis? These sequence will provide a better understanding of the relatedness of LSDV circulating in Africa.

Q2. The clustering of SGI and SGII needs to be explained or referenced. Clustering names (Cluster 1.1 and 1.2) based on Biswas, et al., 2019 are normally used.

Q3. Please check and confirm that the reference numbers are correct. Certain on the references do not speak to the statement that is referenced.

Line 40. It should be “water buffaloes”.

Line 69: It should be “insect”

Line 114: What buffer / solution were used to homogenize the samples?

Lines 197: It should be noted (and stated) that not all the sequences in this group are Neethling-derived vaccines or recombinants from China and Russia. The original Neethling type strain as well as various field isolates also belongs to this group (MN636843).

Author Response

The response to the Reviewer comments are attached

Reviewer 3 Report

The author evaluate poxvirus infections in cattle in Nigeria using a HRM assay to identify specific poxviruses including LSDV and parapoxvirus. The authors have identified both LSDV and PCPV and performed some sequence analysis on the identified viruses. The sequencing data demonstrated that the LSDV circulating were closely related to circulating LSDV in other regions and an additional LSDV were unique in there sequence of the RPO30 gene.

Minor comments.

Line 61 [23, 24].

Line 91 add a .

Line 103 add a .

Line 108 how were the tissues homogenized? mortar and pestle? Please describe.

Line 111 (Bowden et al., 2008) add this as a reference with a number and add the reference to the reference list. Also on line 154.

Author Response

Comments to Reviewer are here attached

Round 2

Reviewer 1 Report

Please find comments in the uploaded document.

Author Response

We have revised and increased the text of the manuscript to more than 4000 words by clarifying the morbidity and mortality rate of this study from Table 1 in the results and discussion, and by addressing the comments by reviewer 1.

Response to Reviewer 1 Comments (Round 2)

Author response:  In addition to the text changes based of the pdf file, we have addressed the comments below.

 Q1. Add sentence on morbidiy here and mention mortality. LSD morbidity is around 10 – 14%, however morbidity of 40% and ven 100% has been reported [  ]. Mortality rate ranges from 1 - 5%... You indicated in mild infections - what about severe infections? It is normally below 10% but can occasionally reach 50 – 75% e.g. Turkey, reaching 54.8% in Holstein cattle (Sevik and Dogan 2017)

Author response: We have revised the paragraph on morbidity and mortality rates and also included the case fatality rate which is normally higher than mortality rate as the example in Turkey.

 Q2. The genus names are Parapoxvirus and Capripoxvirus, Orthopoxvirus - then italics but

parapoxviruses and capripoxviruses.

Author response: We have changed as recommended.

 Q3. Sentence is not clear. They are kept on farms but also move around to graze? Samples were collected from both animals on farms and animals moving across communities. Correct?

Author response: Clarified

 Q4. Two different degree signs are used here. Be consistent with the use of the degree sign and use the correct degree sign. There should not be a space between the number and the degree sign. All other units do have a space except % and °C

Author response: Corrected

 Q3. The presence of any reports that have been published of other co-infections are important.

Author response: We have added more examples of reports on co-infection
